# B$_0$ Correction for 3T Amide Proton Transfer (APT) MRI Using a Simplified Two-Pool Lorentzian Model of Symmetric Water and Asymmetric Solutes

**Yibing Chen** [1,†], **Xujian Dang** [1,†] , **Benqi Zhao** [2], **Zhuozhao Zheng** [2], **Xiaowei He** [1] **and Xiaolei Song** [3,*]

[1] Xi'an Key Laboratory of Radiomics and Intelligent Perception, School of Information Sciences and Technology, Northwest University, Xi'an 710069, China; cyb@stumail.nwu.edu.cn (Y.C.); dangxujian@stumail.nwu.edu.cn (X.D.); hexw@nwu.edu.cn (X.H.)
[2] Department of Radiology, Beijing Tsinghua Changgung Hospital, Beijing 102218, China; zbqa01279@btch.edu.cn (B.Z.); zzza00509@btch.edu.cn (Z.Z.)
[3] Center for Biomedical Imaging Research, Department of Biomedical Engineering, Tsinghua University, Beijing 100084, China
[*] Correspondence: songxl@tsinghua.edu.cn
[†] These authors contributed equally to this work.

**Abstract:** Amide proton transfer (APT)-weighted MRI is a promising molecular imaging technique that has been employed in clinic for detection and grading of brain tumors. MTR$_{asym}$, the quantification method of APT, is easily influenced by B$_0$ inhomogeneity and causes artifacts. Current model-free interpolation methods have enabled moderate B$_0$ correction for middle offsets, but have performed poorly at limbic offsets. To address this shortcoming, we proposed a practical B$_0$ correction approach that is suitable under time-limited sparse acquisition scenarios and for B$_1 \geq 1$ µT under 3T. In this study, this approach employed a simplified Lorentzian model containing only two pools of symmetric water and asymmetric solutes, to describe the Z-spectral shape with wide and 'invisible' CEST peaks. The B$_0$ correction was then performed on the basis of the fitted two-pool Lorentzian lines, instead of using conventional model-free interpolation. The approach was firstly evaluated on densely sampled Z-spectra data by using the spline interpolation of all acquired 16 offsets as the gold standard. When only six offsets were available for B$_0$ correction, our method outperformed conventional methods. In particular, the errors at limbic offsets were significantly reduced ($n = 8$, $p < 0.01$). Secondly, our method was assessed on the six-offset APT data of nine brain tumor patients. Our MTR$_{asym}$ (3.5 ppm), using the two-pool model, displayed a similar contrast to the vendor-provided B$_0$-orrected MTR$_{asym}$ (3.5 ppm). While the vendor failed in correcting B$_0$ at 4.3 and 2.7 ppm for a large portion of voxels, our method enabled well differentiation of B$_0$ artifacts from tumors. In conclusion, the proposed approach could alleviate analysis errors caused by B$_0$ inhomogeneity, which is useful for facilitating the comprehensive metabolic analysis of brain tumors.

**Keywords:** amide proton transfer (APT) MRI; B$_0$ inhomogeneity correction; brain tumors; Lorentzian fitting; chemical exchange saturation transfer (CEST) MRI

## 1. Introduction

As a type of chemical exchange saturation transfer (CEST) imaging [1–3], amide proton transfer (APT) imaging is a promising, non-invasive molecular MRI technique that can detect endogenous mobile proteins and peptides in tissue [4,5]. Numerous institutions worldwide have demonstrated that APT imaging adds important value to the standard clinical MRI sequences in brain tumor diagnoses, such as finding biomarkers, monitoring tumor progression and response to treatment, grading gliomas, etc. [6–9]. Due to the asymmetric nature of CEST signals, asymmetry analysis of the magnetization transfer ratio (i.e., MTR$_{asym}$) is employed to quantify APT MRI, which equates to the subtraction of

normalized saturation signals at two symmetric offsets around the water frequency (i.e., 0 ppm). $MTR_{asym}$ is susceptible to $B_0$ inhomogeneity, and $B_0$ artifacts interfere with the identification and analysis of brain tumors. Therefore, $B_0$ inhomogeneity correction is important for the quantification and clinical applications of APT imaging.

Limited by the scan time, in the clinical APT protocol, a few saturation offsets are acquired around +3.5 ppm and −3.5 ppm for the post-processing of $B_0$ inhomogeneity correction, instead of real-time $B_0$ correction being performed during acquisition [10–13]. The most commonly used post-processing correction methods are interpolation-based methods, which include two steps. First, densely sampled signals with intervals of 0.1 ppm are interpolated from sparsely acquired signals using spline or other interpolated methods [14–18]. Second, the $B_0$ inhomogeneity is corrected through a $B_0$ inhomogeneity ($\Delta B_0$) map of the same image geometry. A $\Delta B_0$ map can be obtained using water saturation shift referencing (WASSR) [19,20], Dixon [21], or Lorentzian-based methods [22].

Interpolation-based methods require high-frequency resolutions and signals close to the water frequency to provide line shape and adequate neighborhood information. For example, Debnath et al. found that linear interpolation was suitable for the $B_0$ correction of APT data ($B_1 = 2$ μT) acquired at 64 offsets (−14~14 ppm with 0.5 ppm intervals) [15]. However, as it is limited by scan time, when using the clinical APT protocol, only a few saturation offsets can be sampled around ±3.5 ppm. Due to insufficient data acquisition, interpolation-based methods perform poorly at limbic offsets, which means the APT protocol typically only provides $MTR_{asym}$ (3.5 ppm). Therefore, the acquisition signals of the APT protocol are not actually fully used, meaning some important metabolites are discarded, such as the fast exchange amine (2.7 ppm) and semi-solid macromolecules (4.3 ppm) [21].

The a priori introduction of a line-shape constraint may compensate for the disadvantages caused by insufficient data acquisition in the $B_0$ correction process. Zhou et al. reported Z-spectral line-shapes of brain tumor patients ($B_0 = 3T$, $B_1 = 2$ μT) in their APT imaging review, which indicates that Z-spectral line-shapes are determined by four effects, i.e., direct water saturation (DS), semi-solid magnetization transfer (MT), CEST and the relayed nuclear Overhauser effect (NOE) [9]. The four effects can be simply divided into two components according to whether the effect is symmetry around water frequency, i.e., a symmetric component including DS, and an asymmetric component including MT, CEST and NOE. Lee et al. proposed a model-based CEST analysis method, which also separated Z-spectra into symmetric and asymmetric parts and used a Lorentzian model to fit the symmetric part [23]. Therefore, limited by few acquisition offsets, a two-pool Lorentzian model may be the correct choice to describe Z-spectral line shapes under 3T with high $B_1$, in which one pool fits the symmetric water and another pool fits all asymmetric solutes.

Herein, we propose a practical $B_0$ correction approach for use in the most popular six-offset acquisition protocol [24]. Using this approach, we employed a two-pool (symmetric water and asymmetric solutes) Lorentzian model to fit the Z-spectral line shape of human brains at 3T with $B_1 = 2$ μT, in a voxel-by-voxel manner. We evaluated our method using in vivo APT data acquired from the brains of healthy volunteers and tumor patients. The contributions of the present study are as follows: (a) In theory, we propose a simplified two-pool Lorentzian model that is suitable to describe the Z-spectral line shape of human brains under 3T with $B_1 \geq 1$ μT. The reduced number of model parameters allowed for fitting using less frequency offsets, i.e., six offsets as in the popular APT protocol. (b) Compared with conventional model-free interpolation, the proposed method could better recover the Z-spectral signals and improve $B_0$ correction performance, especially for limbic offsets.

## 2. Theory

To robustly describe the Z-spectral line shape under 3T with high $B_1$ (2 μT), a simplified two-pool Lorentzian model was chosen as an a priori line shape to fit the spline-interpolated initial Z-spectra (ranging ±2.5~±4.5 ppm). Given the small number of acquired offsets

and the broadening of the peaks, we used the simplest standard to construct our two-pool model, i.e., symmetric water was considered as one pool, and all asymmetric solutes were considered as another pool, which included a large portion of semi-solid macro-molecules, a relayed nuclear Overhauser effect (NOE), amides and other metabolites. Equations (1) and (2) present the model function of our two-pool Lorentzian method.

$$\frac{S_{sat}(\Delta\omega)}{S_0} = Z_{base} - \sum_{i=1}^{2} L_i(\Delta\omega) \tag{1}$$

$$L_i(\Delta\omega) = \frac{A_i}{1 + (\Delta\omega - \Delta_i)^2/(0.5W_i)^2} \tag{2}$$

where $i$ = 1 (symmetric water), 2 (asymmetric solutes); the parameter $Z_{base}$ is used to correct for a constant signal reduction; $L_i$ represents a Lorentzian line with a central offset ($\Delta_i$), peak full width at half maximum (FWHM, $W_i$), and peak amplitude ($A_i$); $S_{sat}(\Delta\omega)/S_0$ is the normalized Z-spectrum. Based on previous studies [25,26] and our experiences, the starting points and boundaries of the fitting parameters are shown in Table 1. The flowchart of the proposed $B_0$ correction procedure is illustrated in Figure 1.

**Table 1.** Starting points and boundaries of the 2-pool Lorentzian fitting parameters.

|       | $\Delta_1$ | $W_1$ | $A_1$ | $\Delta_2$ | $W_2$ | $A_2$ | $Z_{base}$ |
|-------|-----------|-------|-------|-----------|-------|-------|-----------|
| start | 0         | 50    | 25    | −2        | 50    | 0.1   | 0.7       |
| lower | −0.5      | 0     | 0     | −4        | −inf  | −inf  | −inf      |
| upper | 0.5       | 100   | 50    | 4         | +inf  | +inf  | +inf      |

Note: +inf and −inf represent the positive and negative infinity, respectively.

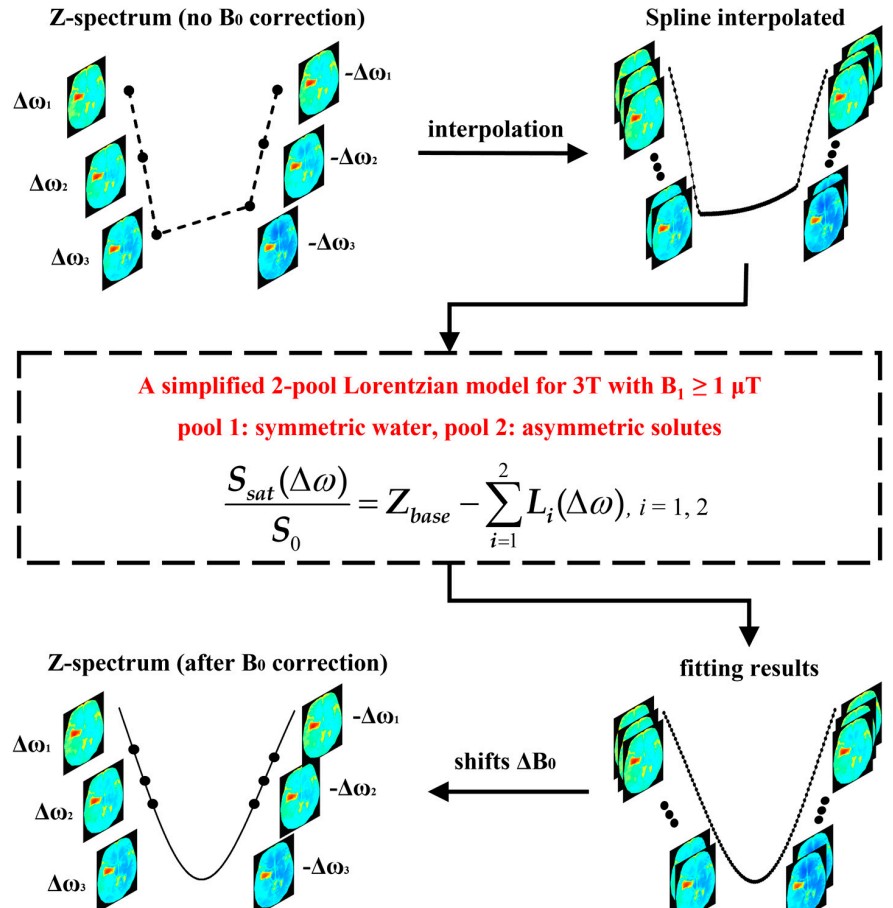

**Figure 1.** The flowchart of the proposed 2-pool-Lorentzian-based $B_0$ correction method.

## 3. Materials and Methods

### 3.1. $B_0$ inhomogeneity Correction

The APT data were normalized by $S_0$ ($-1560$ ppm) without saturation pulse, and smoothed using a $4 \times 4$ median filter according to the acquisition matrix. The proposed 2-pool Lorentzian-based $B_0$ correction method was used in a voxel-by-voxel manner (Figure 1). For each voxel, a 6-offset Z-spectrum was interpolated, generating a fine Z-spectrum with interval of 0.1 ppm. In this study, the cubic-spline method, implemented by MATLAB 2021, a function spline, was employed for interpolation, because it shows less $B_0$ artifacts on MTR$_{\text{asym}}$ (2.7 ppm) than linear and cubic-Hermite interpolation (Figure S1 in the Supplementary Materials). Then, the 2-pool (i.e., symmetric water and asymmetric solutes) Lorentzian model was utilized to fit the line shape of the interpolated Z-spectrum. Finally, after fitting, the Z-spectrum was shifted along the frequency dimension to the correct position, according to a $B_0$ inhomogeneity ($\Delta B_0$).

### 3.2. MTR$_{asym}$ Quantification of APT

After voxel-by-voxel $B_0$ correction, magnetization transfer ratio asymmetry analysis (i.e., MTR$_{\text{asym}}$) was employed to quantify APT, which is defined as Equation (3).

$$\text{MTR}_{\text{asym}}\left(\Delta\omega_j\right) = \frac{S_{sat}\left(-\Delta\omega_j\right) - S_{sat}\left(\Delta\omega_j\right)}{S_0} \tag{3}$$

where $\Delta\omega_j$ represents an offset.

### 3.3. Comparison Methods

The vendor, the cubic-spline interpolation-based method (spline) and the 1-pool Lorentzian-based method (1-pool) were employed as comparison methods. The vendor represents the correction results provided by Philips Healthcare [27]. The 1-pool Lorentzian-based method is similar to the proposed 2-pool Lorentzian-based method, merely replacing the 2-pool (water and solutes) Lorentzian model with the 1-pool (water) Lorentzian model.

### 3.4. Datasets

In this study, first, we recruited 4 healthy volunteers (2 males and 2 females, aged $22 \pm 3.4$ years) and 4 brain tumor patients (3 males and 1 female, aged $53.5 \pm 14.8$ years). From these 8 subjects, densely sampled 16-offset APT data were acquired ($-6.7$, -5.9, $\pm5.1$, $\pm4.3$, $\pm3.5$, $\pm2.7$, $\pm1.9$, $\pm1.1$ and $\pm0.3$ ppm) to validate the accuracy of our method. Then, we recruited 9 brain tumor patients (3 males and 6 females, aged $54.4 \pm 18.6$ years), from whom we acquired 6-offset APT data to enable further comparisons with the vendor. Note that the 16-offset APT protocol was modified by us; therefore, it did not include $B_0$ correction results from the vendor. For our method and comparison methods, 6-offset images ($\pm2.7$, $\pm3.5$ and $\pm4.3$ ppm) were extracted from 16-offset APT data. The $B_0$ correction results using the spline interpolation-based method of 16-offset images were considered to be the gold standard. Of the 13 brain tumor patients, 8 had glioblastomas, 4 had meningiomas and 1 had a metastatic brain tumor from lung cancer.

The study protocol was approved by the institutional review board, and written informed consent was obtained from each subject. MR experiments were performed on a 3T Ingenia MRI system (Philips Healthcare) with a 32-channel phase array coil, using an APT sequence and a turbo spin echo readout. For brain tumor patients, APT data were acquired on the slice centered at the largest areas of the tumors shown on T2w images. The imaging parameters for APT sequences were as follows: $T_{\text{sat}} = 2$ s, $B_1 = 2$ μT, echo time = 8.3 ms, repetition time = 5 s, slice thickness = 7 mm and field-of-view = $220 \times 201$ mm$^2$ with an acquisition voxel size = $2.5 \times 2.5 \times 7$ mm$^3$. $\Delta B_0$ maps were generated using the 3-echo Dixon. Multi-slice T2w images and Gd-T1w images were acquired with a 5 mm slice thickness.

### 3.5. Evaluation Metrics

Using the gold standard as described in Section 3.4, Z-spectra errors and $MTR_{asym}$ errors were employed to evaluate the accuracy of $B_0$ correction, which are defined as Equations (4) and (5).

$$Z\text{-spectra errors} = |Z\text{-spectra(2-pool/comparisons)} - Z\text{-spectra(gold standard)}|, \quad (4)$$

$$MTR_{asym}\text{ errors} = |MTR_{asym}(2\text{-pool/comparisons}) - MTR_{asym}(\text{gold standard})|. \quad (5)$$

One-tailed, paired Student's *t*-tests were used to evaluate the differences between two groups in this study, which were considered to be statistically significant when $p < 0.05$.

## 4. Results

### 4.1. Accuracy Evaluation

Figure 2 shows the $MTR_{asym}$ maps and the corresponding error maps of a representative brain tumor patient. As seen in Figure 2a, before $B_0$ correction, $MTR_{asym}$ maps had severe $B_0$ artifacts, which would have influenced the identification and analysis of tumors (as shown in Figure 2b), especially for 2.7 ppm. The spline-based correction method alleviated some $B_0$ artifacts; however, in the regions with high $B_0$ inhomogeneity ($\Delta B_0 > 0.5$ ppm), the artifacts still appeared on $MTR_{asym}$ (2.7 ppm) and $MTR_{asym}$ (4.3 ppm). The one-pool Lorentzian-based method seemed to eliminate artifacts in regions with high $B_0$ inhomogeneity, but generated wrong $MTR_{asym}$ (4.3 ppm) maps, which were quite different from the gold standard. Compared with the maps generated using the spline and one-pool methods, the three $MTR_{asym}$ maps corrected using the proposed two-pool Lorentzian-based method not only were more similar to the gold standard, but also had fewer $B_0$ artifacts. This can also be validated by the $MTR_{asym}$ error maps, shown in Figure 2c. Using our method, there were fewer $MTR_{asym}$ errors of limbic offsets than when using the spline and one-pool methods, especially in the regions with relatively high $B_0$ inhomogeneity. The $MTR_{asym}$ maps and $MTR_{asym}$ error maps of a representative healthy volunteer are shown in Figure S2 in the Supplementary Materials.

The corresponding region-of-interest (ROI) analyses of the representative tumor patient are shown in Figure 3. Four circle ROIs (radius = 5 voxels) with different $B_0$ inhomogeneities ($\Delta B_0$) are displayed on T2w and a $\Delta B_0$ map (Figure 3a). As seen in Figure 3b, for the ROIs with relatively low $\Delta B_0$ (~0.1 ppm), i.e., ROI 2 and 3, the line shape of two ROIs were similar to each other, showing almost a direct line from 2.7 to 4.3 ppm and an almost 'invisible' peak at −3.5 ppm. Using the corrected Z-spectra with low $\Delta B_0$ as the internal standard, we found that the Z-spectral line-shape of high $\Delta B_0$ (~0.4 ppm) ROIs, corrected using our method, were similar to the internal standard. In contrast, Z-spectra corrected using the spline showed more obvious peaks at 3.5 ppm. Furthermore, Figure 3c shows that the Z-spectra errors using our method were nearly less than 0.5%, and were also less than those generated using the spline and one-pool methods. Similarly, four ROIs, the mean Z-spectra of ROIs and the corresponding Z-spectra errors of the representative healthy volunteer are shown in Figure S3 in the Supplementary Materials.

We statistically analyzed the Z-spectra errors and $MTR_{asym}$ errors of eight subjects (four healthy volunteers and four brain tumor patients). The results are shown in Figure 4, which were consistent with the experimental results from the representative subject (Figures 2 and 3). As seen in Figure 4a, the spline and one-pool methods performed poorly at the limbic offsets, especially for 4.3 ppm, for which the Z-spectra errors of those two methods were around 1%. In contrast, the Z-spectra error from our method for 4.3 ppm was half of that caused by the spline and one-pool methods (~0.5%). For all the offsets, the Z-spectra errors from our method were the lowest among the three correction methods. From Figure 4b, for all the offsets, the $MTR_{asym}$ errors from our method were less than 0.5% and were significantly lower than those from the spline and one-pool methods ($p < 0.01$). In particular, our method dramatically decreased the $MTR_{asym}$ error of 4.3 ppm.

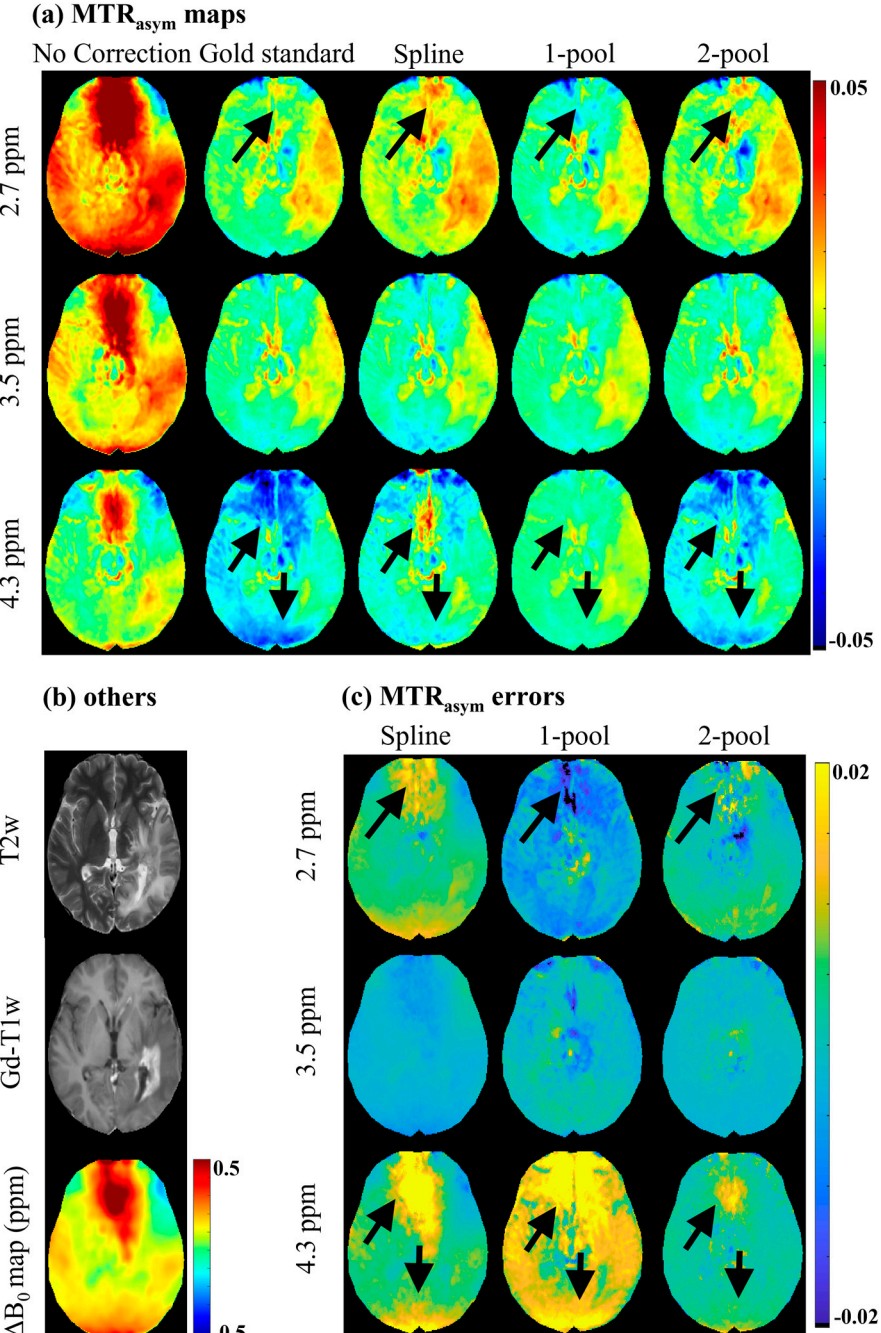

**Figure 2.** MTR$_{asym}$ maps and MTR$_{asym}$ error maps of a representative brain tumor patient. (**a**) MTR$_{asym}$ maps without B$_0$ correction, gold standard, and MTR$_{asym}$ maps corrected using spline, 1-pool Lorentzian and 2-pool Lorentzian methods; (**b**) T2w, Gd-T1w and $\Delta$B$_0$ map; (**c**) the corresponding MTR$_{asym}$ error maps with gold standard.

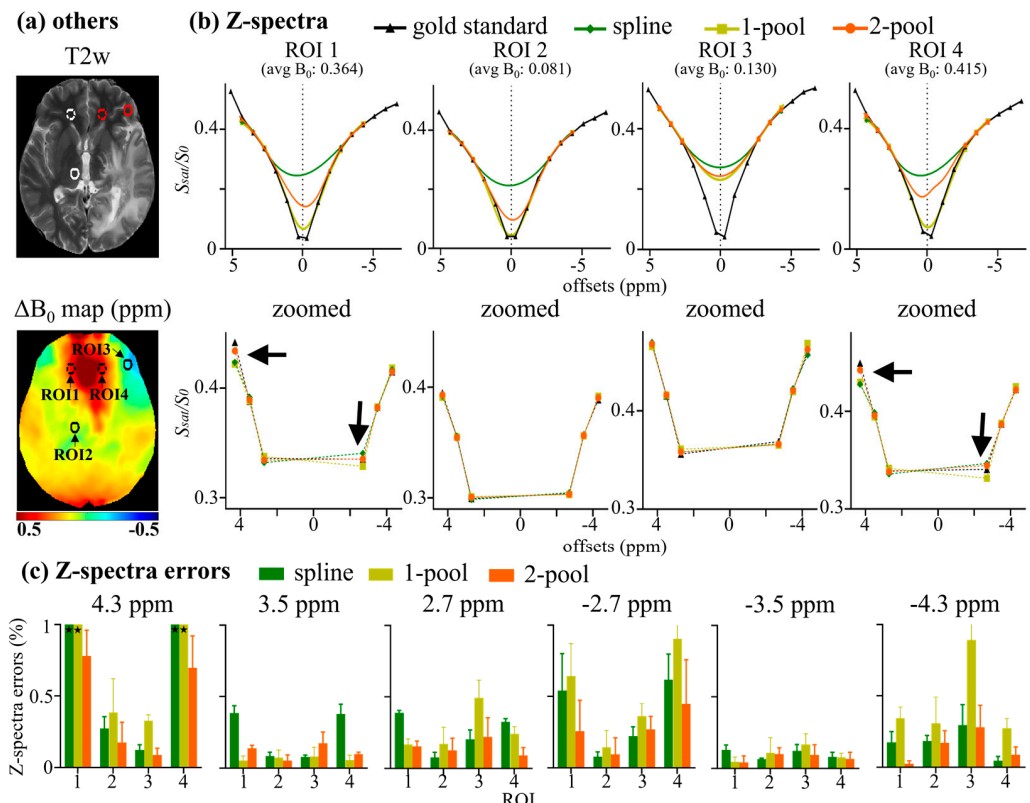

**Figure 3.** ROI analysis of a representative tumor patient. (**a**) Four circle ROIs (radius = 5 voxels) with different $B_0$ inhomogeneity, shown on T2w and $\Delta B_0$ map; (**b**) mean Z-spectra of ROIs, including gold standard, and Z-spectra corrected using spline, 1-pool Lorentzian, and 2-pool Lorentzian methods; (**c**) the corresponding Z-spectra error with gold standard.

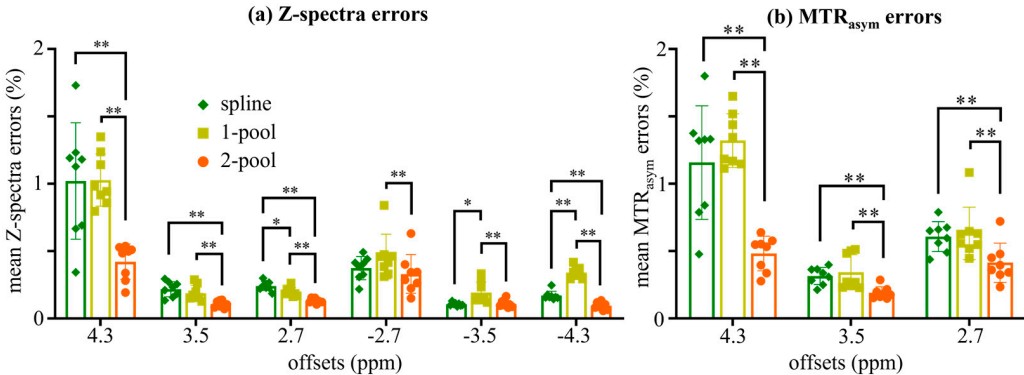

**Figure 4.** Statistical analysis of 8 subjects (4 healthy volunteers and 4 brain tumor patients). (**a**) The statistical results of mean Z-spectra errors corrected using spline, 1-pool Lorentzian and 2-pool Lorentzian methods at 6 offsets. (**b**) The statistical results of mean $MTR_{asym}$ error corrected using spline, 1-pool Lorentzian and 2-pool Lorentzian methods; * $p < 0.05$; ** $p < 0.01$.

As suggested by the experiments with the gold standard, and especially by the statistical results, our method reduced the Z-spectra error and $MTR_{asym}$ error more effectively than the interpolation-based method (i.e., spline) and the one-pool Lorentzian-based method (i.e., one-pool).

### 4.2. Comparison with Vendor

To compare our method with the vendor, the six-offset APT data of nine brain tumor patients were acquired, as described in Section 3.4. Figure 5 displays the $MTR_{asym}$ maps and corresponding ROI analyses of a representative meningioma patient. Four circle ROIs

(radius = 5 voxels) with different $\Delta B_0$ are shown in T2w, Gd-T1w and a $\Delta B_0$ map (Figure 5a). Figure 5b shows $MTR_{asym}$ without $B_0$ correction and $MTR_{asym}$ corrected using the vendor, spline, one-pool Lorentzian and two-pool Lorentzian methods. As seen in Figure 5b, the correction results from the vendor still showed obvious $B_0$ artifacts on $MTR_{asym}$ maps, even for $MTR_{asym}$ (3.5 ppm). Due to the interference of $B_0$ artifacts, we could not identify the tumor region without Gd-T1w and T2w. The one-pool-Lorentzian-based method also had severe $B_0$ artifacts, like the vendor. The spline method and our two-pool-Lorentzian-based method efficiently reduced $B_0$ artifacts, which could help in the identification of tumors. However, the spline-based method still displayed some $B_0$ artifacts at limbic offsets, especially at 4.3 ppm, while our two-pool-Lorentzian-based method also reduced those artifacts. Similar to the method used in Section 4.1, the corrected Z-spectra of low $\Delta B_0$ ROIs (<0.1 ppm) were considered to be the internal standard for comparison with the corrected Z-spectra of high $\Delta B_0$ ROIs (>0.3 ppm). As seen in Figure 5c, the corrected Z-spectral line shapes observed using our method were close to the internal standard, while the use of the spline method caused a peak at 3.5 ppm, and the line shapes observed using the one-pool method were too symmetric.

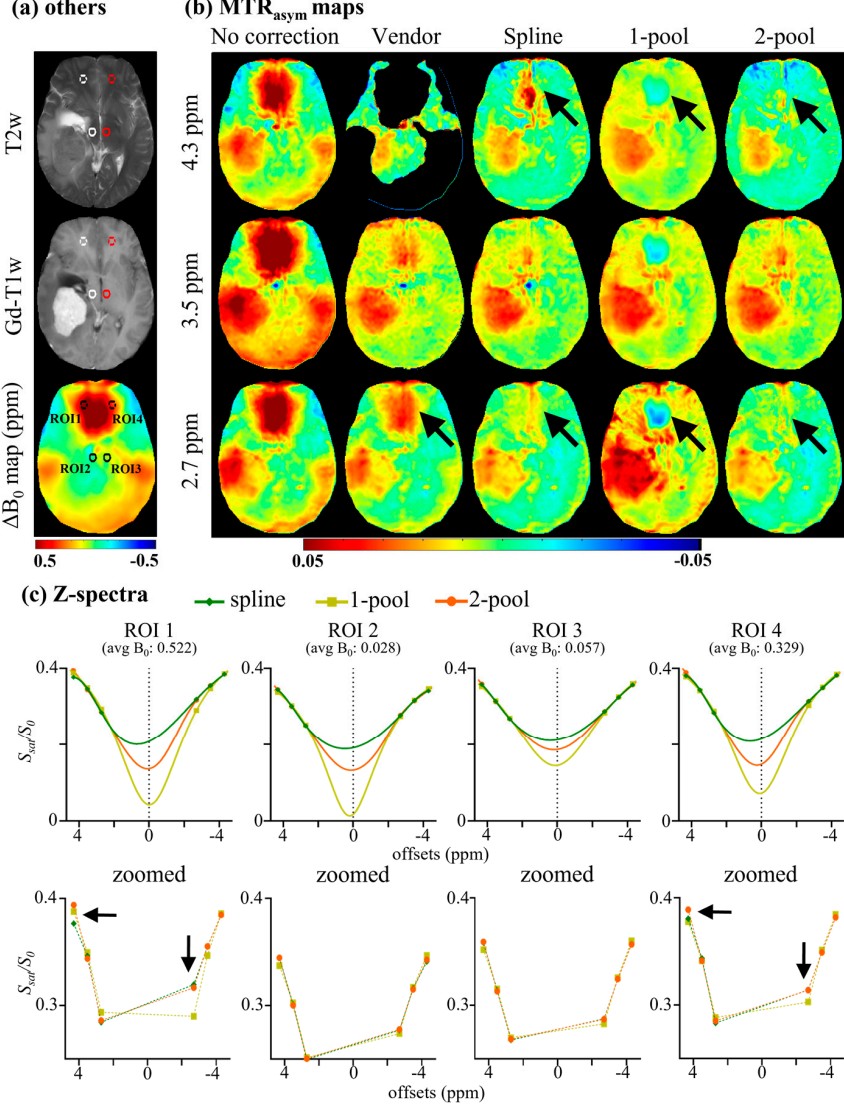

**Figure 5.** $MTR_{asym}$ maps and ROI analysis of a meningioma patient. (**a**) Four circle ROIs (radius = 5 voxels) with different $B_0$ inhomogeneities, shown on T2w, Gd-T1w, and $\Delta B_0$ map; (**b**) $MTR_{asym}$ maps without $B_0$ correction and $MTR_{asym}$ maps corrected using vendor, spline, 1-pool Lorentzian and 2-pool Lorentzian methods; (**c**) mean Z-spectra of ROIs.

To further evaluate the effects of $B_0$ inhomogeneity on tumor analysis, we compared the mean $MTR_{asym}$ values of tumor ROIs and high $\Delta B_0$ ROIs ($|\Delta B_0| > 0.25$ ppm), corrected using the vendor, spline and two-pool Lorentzian methods. Tumor ROIs were annotated on Gd-T1w by one experienced radiologist, and high $\Delta B_0$ ROIs were generated via threshold segmentation with threshold values = 0.25 ppm. Figure 6a shows the tumor ROI (overlapped on Gd-T1w), high $\Delta B_0$ ROI (overlapped on $\Delta B_0$ map) and T2w of another representative patient. As seen in Figures 5b and 6b, $MTR_{asym}$ (4.3 ppm) corrected using the vendor filtered too many voxels; therefore, we excluded it from the statistical analysis. Figure 6c shows that $MTR_{asym}$ corrected using the vendor could not differentiate tumors from $B_0$ artifacts, while the spline method and our two-pool Lorentzian method could efficiently reduce $B_0$ artifacts, which enabled tumors and $B_0$ artifacts to be distinguished ($p < 0.05$). The mean $MTR_{asym}$ values of high $\Delta B_0$ ROIs corrected using our two-pool Lorentzian method were slightly lower than those obtained using the spline method at 2.7 and 4.3 ppm, which may have been due to the reduced $B_0$ artifacts, such as the regions indicated by black arrows in Figures 2, 5b and 6b.

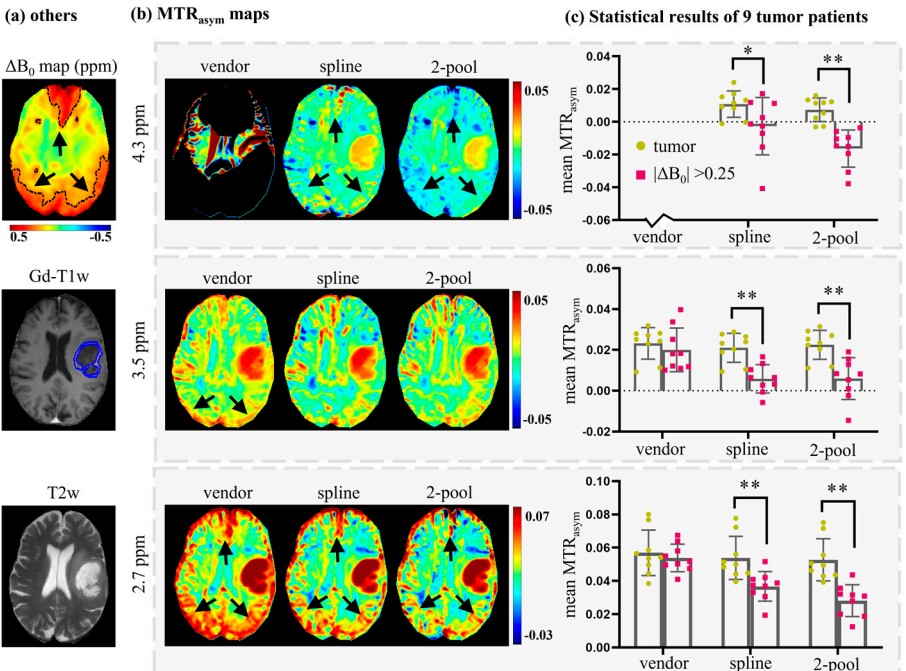

**Figure 6.** A representative brain tumor patient (**a**,**b**) and the statistical results of nine brain tumor patients (**c**). (**a**) ROIs of high $B_0$ inhomogeneity regions (overlapped on $\Delta B_0$ map), ROI of tumor (overlapped on Gd-T1w) and the corresponding T2w. (**b**) $MTR_{asym}$ maps corrected using the vendor, spline, and 2-pool Lorentzian methods. (**c**) The statistical results of nine tumor patients, which compared tumor regions with high $B_0$ inhomogeneity regions on $MTR_{asym}$ corrected using the vendor, spline and 2-pool Lorentzian methods. Because the vendor filtered many voxels of $MTR_{asym}$ (4.3 ppm), we excluded $MTR_{asym}$ (4.3 ppm) provided by the vendor. * $p < 0.05$; ** $p < 0.01$.

## 5. Discussion

In this study, to improve $B_0$ correction performance and fully use all the acquisition offsets, we proposed a practical $B_0$ correction approach for the most popular six-offset acquisition APT protocol. This approach employed a two-pool (symmetric water and asymmetric solutes) Lorentzian line to fit the Z-spectral shape of human brains at 3T with $B_1 = 2$ μT, in a voxel-by-voxel manner (Figure 1). We evaluated our method through two kinds of experiments. Firstly, to validate the accuracy of our method, we acquired densely sampled 16-offset APT data of eight subjects, using its spline-interpolation correction results as the gold standard. As suggested by this experiment, and especially by the statistical results, the use of our method reduced $MTR_{asym}$ errors more efficiently than the spline-

interpolation-based method and one-pool-Lorentzian-based method ($p < 0.01$). For 4.3 ppm, the error of Z-spectra and MTR$_{asym}$ corrected using our method were almost half of the errors caused by the spline and one-pool Lorentzian methods (Figures 2–4). Secondly, for comparison with the vendor, we recruited nine brain tumor patients, from whom we acquired six-offset APT data. The experimental results suggested that our two-pool Lorentzian methods efficiently reduced B$_0$ artifacts, which enabled tumor regions and B$_0$ artifacts to be distinguished ($p < 0.01$), while the vendor could not differentiate tumors from artifacts (Figures 5 and 6).

Interpolation-based methods are very commonly used in B$_0$ correction. Debnath et al. compared the B$_0$ correction performance of different interpolation algorithms with different step sizes on 64-offset APT data (B$_0$ = 3T, B$_1$ = 2 µT), and found that linear interpolation-based methods were suitable [15]. As seen in Figure S1, the performance of different interpolation-based methods (linear, cubic-Hermite, and cubic-spline) were similar when using 16 densely sampled offsets. However, for the sparse acquisition scenario (i.e., six offsets), linear and cubic-Hermite showed severe B$_0$ artifacts on MTR$_{asym}$ (2.7 ppm); cubic-spline (spline) outperformed these two algorithms but still displayed B$_0$ artifacts. This may suggest that we need to choose interpolation algorithms for B$_0$ correction with caution when sampled offsets are few, and spline interpolation may be suitable.

The step size of 6-offset APT data is the same with 16-offset data, but 6-offset acquisition does not cover the entire Z-spectrum. Windschuh et al. and Stancanello et al. also indicated that a full Z-spectrum is important for B$_0$ correction [12,28]. These results may suggest that Z-spectral line shapes are necessary for accurate B$_0$ correction. Zhou et al. reported that the Z-spectral line shape under 3T with high B$_1$ is determined via the symmetric effect around water frequency (i.e., direct water saturation) and asymmetric effects (i.e., MT, NOE and CEST) [9,24]. Lee et al. separated Z-spectra into symmetric and asymmetric components for analysis and used a Lorentzian model to fit the symmetric component [23]. In addition, the broadening Z-spectra under 3T with high power did not show specific "visible" peaks. Therefore, a two-pool (symmetric water and asymmetric solutes) Lorentzian model may accurately describe the Z-spectral line shape, which could provide important a priori line-shape information for the B$_0$ correction of six-offset APT data. As seen in Figure 3, using the Z-spectra of regions with low B$_0$ inhomogeneity as the internal standard, we found that the Z-spectral line shapes corrected using our method were similar to the internal standard, while those corrected using the spline-based method were determined by acquired data, and those corrected using the one-pool-Lorentzian-based method were too symmetric (Figure 3). The statistical analysis of Z-spectra errors in eight subjects also demonstrated that our method was close to the gold standard and reduced errors more effectively than the spline and one-pool Lorentzian methods (Figure 4).

The proposed method could alleviate the analysis errors caused by B$_0$ inhomogeneity and could be combined with CEST analysis methods to provide more metabolic information. For example, in this study, we provided more MTR$_{asym}$ maps with higher image quality and less B$_0$ artifacts than the vendor (Figures 5 and 6), i.e., MTR$_{asym}$ (2.7 ppm) reflecting fast exchange amide, including glutamate and MTR$_{asym}$ (4.3 ppm) reflecting semi-solid MT components. Glutamate is an important energy source for tumor cells, and it always appears in tumor cells that are rapidly growing and dividing [29,30]. In addition, it is a biomarker for the diagnosis and assessment of various psychiatric and neurological disorders [31,32]. MT reflects myelin integrity and, to a lesser extent, cell membrane integrity, which could be used as a biomarker for neurological diseases in which the myelination of the brain is altered, such as in multiple sclerosis [33,34]. Moreover, Mehrabian et al. also found that MT is sensitive to treatment-induced changes in glioblastomas [35].

As the next step, we will combine our method with more CEST analysis methods to provide a comprehensively metabolic delineation of brain tumors. In addition to MTR$_{asym}$, CEST frequency importance analysis could provide more metabolic features of all the acquired offsets, including upfield NOE offsets [36]. We will also combine our B$_0$ correc-

tion method with the CEST frequency importance analysis method to fully use all the acquired offsets.

Although our method showed better performance than the interpolation-based method, some limitations remain. (1) In this study, we employed 17 subjects to demonstrate the performance of our method; however, for clinical application, our findings need to be supported by experimental results from studies that have used a larger number of subjects. We will collect more data to evaluate the proposed method more comprehensively. (2) A drawback of the Lorentzian fitting method is the relatively lengthy computation time required. Due to the voxel-by-voxel correction, the computation time of our method was about 7 min for one subject. (3) The Lorentzian fitting method is sensitive to the starting points and boundaries of parameters, and there is not a common method used to determine appropriate starting points and boundaries. Meanwhile, the choice of parameter boundaries influences the fitting time. (4) Limited by the small number of acquisition offsets, we used a simplified two-pool Lorentzian model to replace the multiple-pool model. Fortunately, some studies have focused on accelerating Lorentzian fitting. For example, Yao et al. classified voxels into several clusters, and only conducted Lorentzian fitting once for a cluster to reduce the fitting time [22]. Zaiss et al. employed neural networks to predict the parameters of Lorentzian function, and accelerated Lorentzian fitting to several seconds [37]. In this study, we demonstrated the feasibility of our method. As the next step, we will combine our method, neural networks and densely sampled simulation data, which will enable us to realize a quick multiple-pool Lorentzian-based correction method without tuning fitting parameters.

## 6. Conclusions

In this study, a practical $B_0$-correction approach is proposed, which employed a simplified two-pool Lorentzian model for Z-spectral fitting with $B_1 \geq 1$ μT under 3T. For both healthy subjects and tumor patients, our approach outperformed conventional interpolation, allowing for better correction at limbic offsets. Therefore, this approach may allow for the efficient extraction of CEST contrast at multiple frequency offsets, and facilitate the more comprehensive metabolic analysis of brain tumors.

**Supplementary Materials:** The following supporting information can be downloaded at: https://www.mdpi.com/article/10.3390/tomography8040165/s1, Figure S1. $MTR_{asym}$ maps corrected employing three interpolation-based methods (i.e., linear, cubic-Hermiter, and cubic-spline), using 16 offsets and 6 offsets, respectively. Figure S2. $MTR_{asym}$ maps and $MTR_{asym}$ error maps of a representative healthy volunteer. (a) $MTR_{asym}$ maps without $B_0$ correction, gold standard and $MTR_{asym}$ maps corrected using spline, 1-pool Lorentzian and 2-pool Lorentzian methods; (b) T2w, Gd-T1w and $\Delta B_0$ map; (c) the corresponding $MTR_{asym}$ error maps with gold standard. Figure S3. ROI analysis of a representative healthy volunteer. (a) Four circle ROIs (radius = 5 voxels) with different $B_0$ inhomogeneities, shown on T2w and $\Delta B_0$ map; (b) mean Z-spectra of ROIs, including gold standard, and Z-spectra corrected using spline, 1-pool Lorentzian, and 2-pool Lorentzian methods; (c) the corresponding Z-spectra errors with gold standard.

**Author Contributions:** Methodology, Y.C., X.D. and X.S.; software, X.D.; formal analysis, Y.C., X.S.; data curation, B.Z., Z.Z.; writing—original draft preparation, Y.C.; writing—review and editing, Y.C., X.D. and X.S.; supervision, X.S., X.H. and Z.Z.; funding acquisition, X.H. and X.S. All authors have read and agreed to the published version of the manuscript.

**Funding:** This research was funded by the National Natural Science Foundation of China, grant numbers 82071914, 61971350; the National Key R&D Program of China, grant number 2019YFC1521101; the Xi'an Science and Technology Plan, grant numbers 201805060ZD11CG44.

**Institutional Review Board Statement:** The study was conducted in accordance with the Declaration of Helsinki, and approved by the Institutional Review Board of Beijing Tsinghua Changgung Hospital and school of Medicine, Tsinghua University.

**Informed Consent Statement:** Written informed consent was obtained from the subjects to publish this paper.

**Data Availability Statement:** The human data used in this study cannot be shared at this time as the data also form part of an ongoing study.

**Acknowledgments:** The authors are grateful to Changhao Zhu from Northwest University for his generous help with data processing.

**Conflicts of Interest:** The authors declare no conflict of interest.

## Abbreviations

| | |
|---|---|
| APT | amide proton transfer |
| $MTR_{asym}$ | asymmetry analysis of magnetization transfer ratio |
| CEST | chemical exchange saturation transfer |
| $\Delta B_0$ | $B_0$ inhomogeneity |
| WASSR | water saturation shift referencing |
| DS | direct water saturation |
| MT | magnetization transfer |
| NOE | nuclear Overhauser effect |
| $\Delta_i$ | central offset of the Lorentzian line |
| FWHM | peak full width at half maximum |
| $A_i$ | peak amplitude |
| ROI | region-of-interest |

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
