# Peer review of "B0 Correction for 3T Amide Proton Transfer (APT) MRI Using a Simplified Two-Pool Lorentzian Model of Symmetric Water and Asymmetric Solutes"

_tomography, doi:10.3390/tomography8040165_

Round 1

Reviewer 1 Report

why recruit fewer patients into the control group when it carries more statistical weight? 

Reviewer 2 Report

Overall, I find the paper quite interesting, and may I congratulate the authors for their scientific contribution. However, some corrections are still needed in order to publish this work. First, a native speaker should read this work, as there are many grammatical (e.g. line 19 "vender", line 51 "...would bring additional asymmetric", line 200: error of our method were...") and syntactical errors (e.g. lines 79-82, line 144 "paried") in it. Pls, correct this. Pls, avoid redundancies and combining past and present tense throughout the manuscript.

Moreover, the introduction is very bumpy and especially the first lines (34-43) are difficult to read, so a revision of this text passage is needed to get a good start for all readers in advance. In my opinion, the APT CEST method used should be mentioned in the first line and then built upon.

Furthermore, the 2-pool Lorentzian fitting was used for healthy individuals as well, pls indicate this in your title, also identify the study design in your title.

Pls, include the study design and the number of subjects in your abstract. Pls, also include more results in the abstract, this may be underrepresented compared to the description of the methods used. Pls, correct this. Furthermore, the method may reduce or minimizer errors, but not achieve lower errors. What are lower errors? Pls, correct this? Pls, include p- value. Stating that your method "may facilitate more comprehensive metabolic analysis of brain tumors", have you achieved this? Pls, if you could include this in your discussion.

To methods: Your study recruited 4 healthy volunteers and 6 tumor patients, but only four tumor patients were included. Pls, describe the exclusion criteria. In addition, pls, include tumor diagnosis here. Pls, describe the comparison method "vendor" further. Who is the supplier?

To discussion: With a mean age of 50 years, tumor patients were considerably older than healthy patients. Pls, elaborate on age-dependent MRT differences, especially for the APT- CEST method with respect to water frequency.

Authors should discuss the results and how they can be interpreted in perspective of previous studies. However, with 4 references, this section is unusually short. Pls, correct this. 

Further, avoid the term "ours" (e.g. 141), and use a scientific term. Furthermore, tables with captions and abbreviations used therein are advantageous.

To conclusions: This section needs further

To references: The reference list should be described following the instructions for authors (e.g. journal abbreviation). Pls, correct this accordingly.

Once again, I would like to appreciate this work and thank the authors for their contribution.

Reviewer 3 Report

1)Abstract: Amide proton transfer (APT) MRI is a promising molecular imaging technique that has  been proved useful for assessing histological grades of brain tumors. APT contrast is quantified by  the asymmetric analysis with respect to water frequency, i.e. MTRasym(3.5 ppm), which requires acquisition of a few saturation offsets around +3.5 ppm and -3.5 ppm for correcting B0 inhomogeneity. However, only MTRasym(3.5 ppm) is provided by vender since conventional interpolation-based cor rection performs poor at limbic offsets. To improve the B0 correction performance at limbic offsets, we propose a practical B0 correction approach for the most popular 6-offset APT acquisition protocol. This approach employed a 2-pool (water and solutes) Lorentzian line to fit the Z-spectral shape of human brain at 3T with B1 = 2 μT, in a voxel-by-voxel manner. Imaging studies on both healthy  volunteers and brain tumor patients were performed, in which densely-sampled 16 offsets were collected with its spline-fitting considered as golden standard. Compared with spline interpolation  and 1-pool (water) Lorentzian fitting, our method achieved significant lower errors, especially for MTRasym(4.3 ppm) and MTRasym(2.7 ppm). In conclusion, the proposed B0-correction method could  produce MTRasym images at multiple acquired offsets, which may facilitate more comprehensive  metabolic analysis of brain tumors. The abstract is quite rumbling and difficult to read. Could you please divide it in different sections (i.e. background, aim, ..)

2) Introduction. L70-75. Herein, to improve the B0 correction performance of limbic offsets and fully utilize 70 all acquisition offsets, we propose a practical B0 correction approach for the most popular  6-offset acquisition protocol [21]. Limited by few acquisition offsets, this approach employed a simplified 2-pool Lorentzian model (i.e., water and solutes) to fit the Z-spectral  shape of human brain at 3T with B1 = 2 μT, in a voxel-by-voxel manner. We have evaluated  our method using in vivo APT data acquired from the brains of healthy volunteers and  tumor patients. Please improve the description of the aim of the study and underline the novelty of the study.

3) L120-123. 3.4. Datasets  We recruited 4 healthy volunteers (2 males and 2 females, aged 22 ± 3.4 years) and 6  tumor patients (3 males and 3 females, aged 50 ± 17.9 years). To validate the accuracy of  our method, all healthy volunteers and four tumor patients acquired densely-sampled 16-  offset APT data (-6.7, -5.9, ±5.1, ±4.3, ±3.5, ±2.7, ±1.9, ±1.1, ±0.3 ppm).  Do you have any comments regarding the low number of the study sample? Could you please underline this limit in the discussion?

4) 4. Results L152-230  Please underline in the text  the most important statistical values to support the results.

5) 6. Conclusions L288-294. In this study, we have proposed a 2-pool (water and solutes) Lorentzian-based  method for B0 correction of the most popular 6-offset APT protocol. By introducing a proper Z-spectral line-shape priori under 3T condition with B1=2 μT, the proposed method  outperformed interpolation-based methods and 1-pool (water) Lorentzian-based method, dramatically alleviating B0 artifacts of MTRasym maps at limbic offsets. This would help  APT protocol providing quantification results at multiple offsets, which may facilitate  more comprehensive metabolic analysis of brain tumors.Please underline the novelty of the paper and the clinical implication of the study.
